# Underwater Image Restoration via Non-Convex Non-Smooth Variation and Thermal Exchange Optimization

**Qingliang Jiao, Ming Liu \*, Pengyu Li, Liquan Dong, Mei Hui, Lingqin Kong and Yuejin Zhao**

Beijing Key Laboratory for Precision Optoelectronic Measurement Instrument and Technology, School of Optics and Photonics, Beijing Institute of Technology, Beijing 100081, China; jiao2006@bit.edu.cn (Q.J.); lipyhebut@163.com (P.L.); kylind@bit.edu.cn (L.D.); huim@bit.edu.cn (M.H.); konglingqin3025@bit.edu.cn (L.K.); yjzhao@bit.edu.cn (Y.Z.)
\* Correspondence: bit411liu@bit.edu.cn

**Abstract:** The quality of underwater images is an important problem for resource detection. However, the light scattering and plankton in water can impact the quality of underwater images. In this paper, a novel underwater image restoration based on non-convex, non-smooth variation and thermal exchange optimization is proposed. Firstly, the underwater dark channel prior is used to estimate the rough transmission map. Secondly, the rough transmission map is refined by the proposed adaptive non-convex non-smooth variation. Then, Thermal Exchange Optimization is applied to compensate for the red channel of underwater images. Finally, the restored image can be estimated via the image formation model. The results show that the proposed algorithm can output high-quality images, according to qualitative and quantitative analysis.

**Keywords:** underwater image restoration; underwater DCP (dark channel prior); non-convex non-smooth variation; adapt parameter selection; thermal exchange optimization

## 1. Introduction

With exploration and research in some underwater fields, such as oceans, lakes, and rivers, the application of underwater detection is more extensive, and the requirements for the quality of the captured images are much higher. However, due to the attenuation characteristics of light propagation in the water, as well as the absorption and scattering effects of particles and impurities in water, the underwater images can be impacted by serious degradation effects, such as atomization, low contrast, noise, blur, and color distortion, which limits their applications in follow-up image processing tasks (such as image recognition and target detection) [1–3]. Therefore, the use of image processing technology to improve the quality of underwater degraded images has become a research focus in recent years.

In early research, the issue of improving underwater image quality was approached by the use of additional information, such as using multiple images [4] or polarization filters [5,6]. These methods can improve the quality of underwater images, but they also have several shortcomings, such as the high cost involved.

In recent years, with the rapid development of computer technology, many underwater image restoration algorithms without additional hardware have been proposed. According to the principles of these algorithms, they can be divided into three categories: the image formation model-based (IFM-based) method, the image formation model-free (IFM-free) method, and the deep-learning-based method.

The IFM-free methods, such as Retinex [7], Gray-World algorithm [8], and multi-scale fusion [9,10], are also called image enhancement methods, which improve the contrast without using underwater imaging models. These methods have high contrast and fast calculation speeds. However, they may amplify noise, which reduces the underwater

image quality and seriously impacts the applications of underwater resource detection and target recognition.

Artificial intelligence technology has made great advancements in recent years, and underwater image restoration methods based on deep learning have also made continuous progress. Park et al. proposed an Adaptive Weighted Multi-Discriminator CycleGAN, which obtained high-quality underwater images [11]. Wang et al. designed a Parallel Convolutional Neural Network (CNN) that included a transmission estimation network and a global ambient light estimation network [12]. Liu et al. proposed an underwater image enhancement method with a deep residual framework, which was composed by cycle-consistent adversarial networks and the deep super-resolution reconstruction model [13]. Li et al. applied a synthesis underwater image to propose a CNN based on underwater scene prior [14]. Recently, unsupervised and self-supervised learning have become a focus in deep learning, and these methods have achieved excellent results in underwater image restoration. Ye et al. built an unsupervised adaptation network to estimate the depth map and restored image [15]. Contrastive learning and generative adversarial networks were utilized by Han et al. to maximize the mutual information between datasets [16]. According to the above analysis, the methods based on deep learning can output high-quality images. The performance of deep learning models depends on the training data. However, obtaining high-quality training data is difficult, especially for underwater images.

The principle of IFM-based methods is based on underwater optical imaging models and prior knowledge of underwater scenes. The most famous method is dark channel prior (DCP) [17], which is not only successfully applied to image defogging, but also effective for underwater image restoration. Berman et al. proposed a non-local prior to estimate haze-free images, which had linear complexity, that requires no training [18]. Peng et al. considered the problem of image blurriness and light absorption, and proposed a depth estimation method to obtain restored images [19]. Zhang et al. improved the underwater image formation model, which estimated the medium transmissions of three channels of an underwater image via joint prior [20]. The attenuation curves prior was applied to estimate transmission maps by Dai et al., who also proposed a color balance algorithm to estimate restored images with a natural appearance [21].

Unlike the influence of haze on light propagation, the influence of water on light propagation is related to the wavelength of light. In general, the longer the wavelength of light, the stronger the absorption and scattering of light by water. Therefore, it is important to compensate for the intensity of the red channel. Azmi et al. used the blue and green channels to estimate the red channel, and particle swarm optimization was used to perform contrast stretching [22]. Zhao et al. studied the optical properties of water, and analyzed the relationship between the RGB channels in underwater images [23]. Galdran et al. employed the method of red channel compensation based on DCP, which realized color correction and visibility improvements [24].

In addition to red channel compensation, the optimization of transmission maps is also an important research direction of IFM-based methods, which improves the contrast and corrects the color of restored images. For the optimization of transmission maps, which can be performed by the method of guided filtering [25,26], variation is a more ideal choice, because it is not only sensitive to the structure and texture of the image, but can also reconstruct the image. Hou et al. designed nonlocal variational [27], and curvature variation regularization [28], which proved that the variational model outperforms the state-of-the-art methods on real-world underwater images.

However, the variational models mentioned above are all convex. It is well known that non-convex norms are more suitable for measuring sparsity than the corresponding convex norms since they are much closer to the $l_0$ norm, which is the exact measure of sparsity, than their convex counterparts [29–31]. The non-convex regularization model was first proposed by Geman et al. [32], and then several non-convex variational models were designed, such as non-convex high-order variation [33] and non-convex hybrid total variation [34]. These

models have shown the superiority of the non-convex norm, as they can preserve weak edges and textures, which is important for underwater target identification and tracking.

In this paper, we propose a novel underwater image restoration algorithm based on DCP and non-convex and non-smooth variation to improve the underwater image quality and correct color. The main contributions of this paper are summarized as follows:

- A transmission map optimization model combining non-convex and non-smooth is proposed;
- A method for adaptively selecting regularization parameters is proposed, which can update parameters while optimizing transmission maps;
- An iteratively reweighted $l_1$ (IRL1) algorithm and alternate multiplier method (ADMM) are used to solve the proposed non-convex and non-smooth model quickly and accurately;
- A transmission map and the background light of the red channel compensation method is designed based on Thermal Exchange Optimization. Because the attenuation of absorption and scattering for red light is the fastest, there are errors when using the fastest attenuating channel (red channel) to estimate the transmission maps of other channels.

## 2. Materials and Methods

According to the underwater imaging mechanism, a novel underwater image restoration method is proposed. Firstly, the Underwater DCP is used to obtain the rough transmission map. Then, the adaptive non-convex and non-smooth variational is used to optimize the green and blue channels. Next, the Thermal Exchange Optimization is used to estimate the transmission map and background light of the red channel via the optimized green and blue channels. Finally, the simplified underwater imaging model is used to obtain the restored image. The algorithm flow is shown in Figure 1.

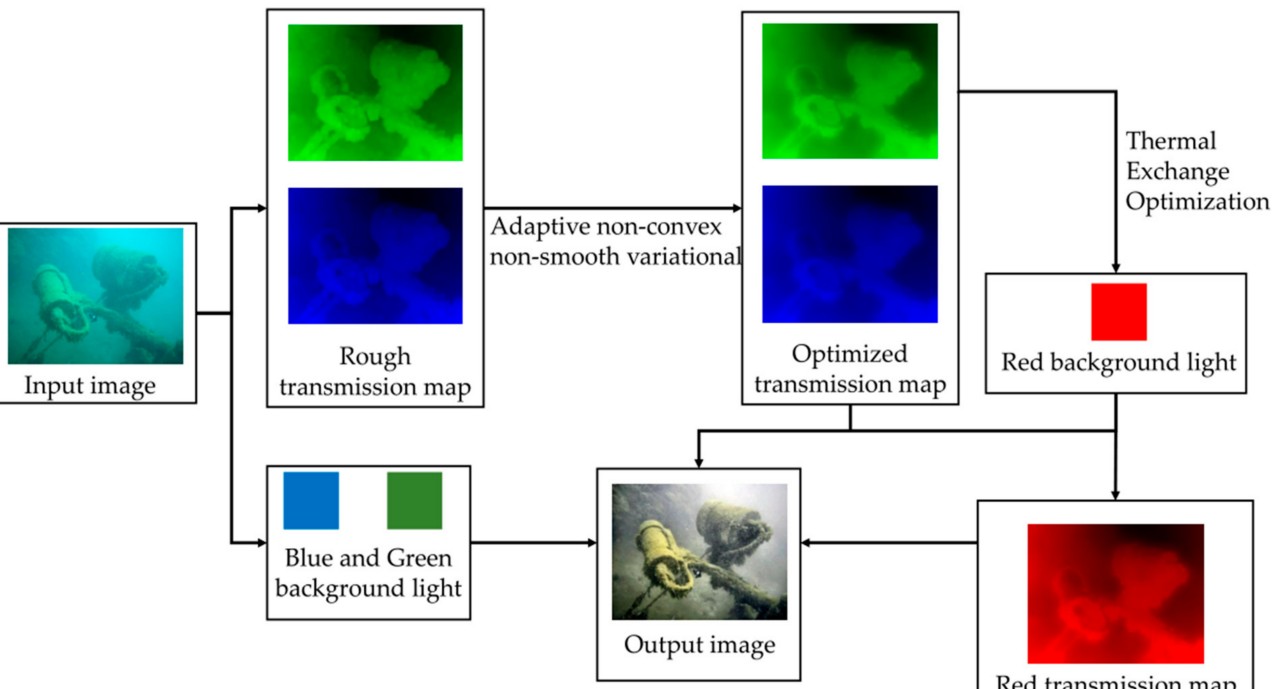

**Figure 1.** Details of the proposed underwater image restoration algorithm.

## 2.1. Underwater imaging and Dark Channel Prior

The attenuation of light in water is a complicated process. To reduce the complexity of the model, a simplified underwater imaging model [35] is used to approximately describe this process. The simplified model can be shown as

$$I(x) = E_D(x) + E_F(x) + E_B(x) \tag{1}$$

where $I$ is the captured image, $E_D$ is direct illumination, $E_F$ represents the forward-scattering of light, $E_B$ is the back-scattering when light is spread underwater, and $x$ is the image coordinate. The influence of $E_F$ in underwater imaging is usually ignored. Therefore, (1) can be rewritten as

$$I(x) = J(x)t(x) + B(1 - t(x)) \tag{2}$$

where $J$ is the ideal underwater image, $t$ is the transmission map, and $B$ is the background light. From (2), the key in underwater image restoration is to estimate the transmission map and background light. DCP is a powerful tool for estimating the transmission map and background light, because there is at least one pixel, which is almost 0, of RGB channels in each local patch. Pan et al. proved mathematically that the sparsity of the dark channel pixels is an inherent property of the blur image [36]. The dark channel can be described as

$$I_{dark}(x) = \min_{y \in \Omega} \left\{ \min_c I^c(y) \right\} = 0, \qquad c \in \{R, G, B\} \tag{3}$$

According to (3), (2) can be rewritten as

$$\min_{y \in \Omega} \left\{ \min_c \frac{I^c(y)}{B^c} \right\} = \min_{y \in \Omega} \left\{ \min_c J^c(y) \right\} + 1 - t(x), \qquad c \in \{R, G, B\} \tag{4}$$

In DCP, the transmission map is defined as

$$t(x) = 1 - \min_{y \in \Omega} \left\{ \min_c \frac{I^c(y)}{B^c} \right\}, \qquad c \in \{R, G, B\} \tag{5}$$

## 2.2. Thermal Exchange Optimization

Thermal Exchange Optimization (TEO) [37] is an intelligent optimization algorithm based on Newton's law of cooling, which has been applied to image segmentation [38], and image fusion [39]. In TEO, a number of agents are set, and then some of them are used as the object temperature and others as the environment temperature. The environment temperature can be updated as

$$T^{new} = T^{env} + \left( T^{old} - T^{env} \right) \exp\left( -\beta \frac{l}{L} \right) \tag{6}$$

$$\beta = \frac{\text{cost(object)}}{\text{cost(worst object)}} \tag{7}$$

$$T^{env} = \left( 1 - \left( c_1 + c_2 \left( 1 - \frac{l}{L} \right) \right) \cdot random \right) T^{env\prime} \tag{8}$$

where $c_1$, $c_2$ are the controlling variables, $T^{old}$ and $T^{new}$ are the previous and current temperatures of the object, $l$ and $L$ are the current and max iteration numbers, respectively. The TEO is conceptually simple and relatively easy to implement, making it an ideal tool to estimate the red channel transmission map and background light.

### 2.3. Adaptive non-convex and non-smooth variational models

In this part, the optimization model of the transmission map is proposed. In this model, the negative Sobolev space $H^{-1}(\Omega)$ is used to protect weak edges and textures. The model is defined as

$$
\begin{aligned}
t &= \underset{t,J}{\operatorname{argmin}} \|\varphi(|\nabla t|)\|_1 + \tfrac{\lambda_1}{2}\left\|\nabla\left(\Delta^{-1}(I - J{\cdot}t - B(1-t))\right)\right\|_2^2 + \tfrac{\lambda_2}{2}\left\|\nabla\left(\Delta^{-1}(J - I_{CLAHE})\right)\right\|_2^2 \\
\left(t^{k+1}, J^{k+1}\right) &= \underset{t^k,J^k}{\operatorname{argmin}} \nabla\varphi\left(\left|\nabla t^k\right|\right)\left\|\nabla t^k\right\|_1 + \tfrac{\lambda_1^k}{2}\left\|\nabla\left(\Delta^{-1}\left(I - J^k{\cdot}t^k - B\left(1-t^k\right)\right)\right)\right\|_2^2 + \tfrac{\lambda_2^k}{2}\left\|\nabla\left(\Delta^{-1}\left(J^k - I_{CLAHE}\right)\right)\right\|_2^2
\end{aligned}
\tag{9}
$$

where $I_{CLAHE}$ is the enhanced origin underwater image via CLAHE [33], $\nabla$ is the gradient operator, $\Delta^{-1}$ is the inverse Laplace operator, and $\varphi(\bullet)$ is a non-convex and non-smooth function. In this paper, the $\varphi(\bullet)$ function is defined as

$$
\varphi(|\nabla t|) = \frac{1}{\alpha}\log(1 + \alpha|\nabla t|), \ \alpha > 0
\tag{10}
$$

From (9), there are two regularized parameters. To improve the generality of the model, the adaptive selection method of regularization parameters is as follows

$$
\begin{aligned}
(t, \lambda_1, \lambda_2) = \ \ &\underset{t,\lambda_1,\lambda_2}{\operatorname{argmin}} \|\varphi(|\nabla t|)\|_1 + \tfrac{\lambda_1}{2}\left\|\nabla\left(\Delta^{-1}(I - J{\cdot}t - B(1-t))\right)\right\|_2^2 \\
&+ \tfrac{\lambda_2}{2}\left\|\nabla\left(\Delta^{-1}(J - I_{CLAHE})\right)\right\|_2^2 \\
&+ \tfrac{a_1}{2}\|\lambda_1 - b\|_2^2 + \tfrac{a_2}{2}\|\lambda_2 - b\|_2^2
\end{aligned}
\tag{11}
$$

where $a_1$, $a_2$, and $b$ are the constants. From (11), the adaptive selection method regularization parameters consist of (9) and the item of parameters. In the calculation, the regularization parameters are updated as the transmission map is updated, which means the regularization parameters are always estimated by the optimizing transmission map, rather than the rough transmission map. This is the most significant difference and advantage of the proposed method compared to the traditional parameter selection method.

### 2.4. IRL1 and ADMM for proposed models

In this section, the IRL1 and ADMM are used to solve (11). First, (11) is divided into three subproblems, the transmission map subproblem and the regularization parameters' subproblems, which can be defined as

$$
\begin{cases}
\left(t^{k+1}, J^{k+1}\right) = \underset{t^k,J^k}{\operatorname{argmin}} \left\|\varphi\left(\left|\nabla t^k\right|\right)\right\|_1 + \tfrac{\lambda_1^k}{2}\left\|\nabla\left(\Delta^{-1}\left(I - J^k{\cdot}t^k - B\left(1-t^k\right)\right)\right)\right\|_2^2 + \tfrac{\lambda_2^k}{2}\left\|\nabla\left(\Delta^{-1}\left(J^k - I_{CLAHE}\right)\right)\right\|_2^2 \\
\lambda_1^{k+1} = \underset{\lambda_1^k}{\operatorname{argmin}} \tfrac{\lambda_1^k}{2}\left\|\nabla\left(\Delta^{-1}\left(I - J^{k+1}{\cdot}t^{k+1} - B\left(1-t^{k+1}\right)\right)\right)\right\|_2^2 + \tfrac{a_1}{2}\left\|\lambda_1^k - b\right\|_2^2 \\
\lambda_2^{k+1} = \underset{\lambda_2^k}{\operatorname{argmin}} \tfrac{\lambda_2^k}{2}\left\|\nabla\left(\Delta^{-1}\left(J^{k+1} - I_{CLAHE}\right)\right)\right\|_2^2 + \tfrac{a_2}{2}\left\|\lambda_2^k - b\right\|_2^2
\end{cases}
\tag{12}
$$

where $k$ is the current iteration number. From (12), the transmission map subproblem is non-convex and non-smooth, which can be solved via IRL1 and ADMM. The regularization parameters' subproblems are convex; their solution can be written as

$$
\begin{cases}
\lambda_1^{k+1} = \dfrac{a_1 b - \left\|\nabla\left(\Delta^{-1}\left(I - J^{k+1}{\cdot}t^{k+1} - B\left(1-t^{k+1}\right)\right)\right)\right\|_2^2}{a_1} \\
\lambda_2^{k+1} = \dfrac{a_2 b - \left\|\nabla\left(\Delta^{-1}\left(J^{k+1} - I_{CLAHE}\right)\right)\right\|_2^2}{a_2}
\end{cases}
\tag{13}
$$

Next, the transmission map subproblem, which is the proposed non-convex non-smooth model (9), is solved by IRL1. Firstly, (9) should be converted to a convex term:

$$\left(t^{k+1}, J^{k+1}\right) = \underset{t^k, J^k}{\operatorname{argmin}} \nabla \varphi\left(\left|\nabla t^k\right|\right)\left\|\nabla t^k\right\|_1 + \frac{\lambda_1^k}{2}\left\|\nabla\left(\Delta^{-1}\left(I - J^k \cdot t^k - B\left(1 - t^k\right)\right)\right)\right\|_2^2 + \frac{\lambda_2^k}{2}\left\|\nabla\left(\Delta^{-1}\left(J^k - I_{CLAHE}\right)\right)\right\|_2^2 \quad (14)$$

From (14), the non-convex problem is converted to a TV-minimizing problem, whose parameter is $\nabla \varphi\left(\left|\nabla t^k\right|\right)$. The parameter is a monotonically decreasing function with respect to $\left|\nabla t^k\right|$. Therefore, (14) cannot penalize the edges, which preserves weak edges and textures effectively.

Then, to solve the non-smooth problem (14), the ADMM is used. After adding auxiliary variables to (14), its Euler–Lagrangian form is described as

$$\left(v^{k+1}, w^{k+1}\right) = \underset{v^k, w^k}{\operatorname{argmin}} \nabla \varphi\left(\left|\nabla t^k\right|\right)\left\|v^k\right\|_1 + \frac{\lambda_1^k}{2}\left\|\nabla\left(\Delta^{-1}\left(I - w^k \cdot t^k - B\left(1 - t^k\right)\right)\right)\right\|_2^2 + \frac{\lambda_2^k}{2}\left\|\nabla\left(\Delta^{-1}\left(w^k - I_{CLAHE}\right)\right)\right\|_2^2 \quad (15)$$

$$s.t. \quad \nabla t = v, \quad J = w$$

$$\left(t^{k+1}, J^{k+1}, v^{k+1}, w^{k+1}\right) = \underset{t^k, J^k, v^k, w^{k+1}}{\operatorname{argmin}} \nabla \varphi\left(\left|\nabla t^k\right|\right)\left\|\nabla v^k\right\|_1 + \frac{\lambda_1^k}{2}\left\|\nabla\left(\Delta^{-1}\left(I - w^k \cdot t^k - B\left(1 - t^k\right)\right)\right)\right\|_2^2$$
$$+ \frac{\lambda_2^k}{2}\left\|\nabla\left(\Delta^{-1}\left(w^k - I_{CLAHE}\right)\right)\right\|_2^2 + \left(s_1^k\right)^{\mathrm{T}}\left(\nabla t^k - v^k\right) + \left(s_2^k\right)^{\mathrm{T}}\left(J^k - w^k\right) \quad (16)$$
$$+ \frac{\sigma_1}{2}\left\|\nabla t^k - v^k\right\|_2^2 + \frac{\sigma_2}{2}\left\|J^k - w^k\right\|_2^2$$

where $s_1$ and $s_2$ are the Lagrangian multipliers, and $\sigma_1$ and $\sigma_2$ are positive constants, which measure the quadratic penalization. Then, (16) can be rewritten as:

$$\left(t^{k+1}, J^{k+1}, v^{k+1}, w^{k+1}\right) = \underset{t^k, J^k, v^k, w^{k+1}}{\operatorname{argmin}} \nabla \varphi\left(\left|\nabla t^k\right|\right)\left\|v^k\right\|_1 + \frac{\lambda_1^k}{2}\left\|\nabla\left(\Delta^{-1}\left(I - w^k \cdot t^k - B\left(1 - t^k\right)\right)\right)\right\|_2^2$$
$$+ \frac{\lambda_2^k}{2}\left\|\nabla\left(\Delta^{-1}\left(w^k - I_{CLAHE}\right)\right)\right\|_2^2 + \frac{\sigma_1}{2}\left\|\nabla t^k - v^k + \frac{s_1^k}{\sigma_1}\right\|_2^2 + \frac{\sigma_2}{2}\left\|J^k - w^k + \frac{s_2^k}{\sigma_2}\right\|_2^2 \quad (17)$$

Then, a variable can be minimized or other variables can be fixed, alternatively. Equation (17) can be decomposed into six subproblems, which can be shown as follows:

$$v^{k+1} = \underset{v^k}{\operatorname{argmin}} \nabla \varphi\left(\left|\nabla t^k\right|\right)\left\|v^k\right\|_1 + \frac{\sigma_1}{2}\left\|\nabla t^k - v^k + \frac{s_1^k}{\sigma_1}\right\|_2^2 \quad (18)$$

$$w^{k+1} = \underset{w^k}{\operatorname{argmin}} \frac{\lambda_1^k}{2}\left\|\nabla\left(\Delta^{-1}\left(I - w^k \cdot t^k - B\left(1 - t^k\right)\right)\right)\right\|_2^2 + \frac{\lambda_2^k}{2}\left\|\nabla\left(\Delta^{-1}\left(w^k - I_{CLAHE}\right)\right)\right\|_2^2 + \frac{\sigma_2}{2}\left\|J^k - w^k + \frac{s_2^k}{\sigma_2}\right\|_2^2 \quad (19)$$

$$t^{k+1} = \underset{t^k}{\operatorname{argmin}} \frac{\lambda_1^k}{2}\left\|\nabla\left(\Delta^{-1}\left(I - w^{k+1} \cdot t^k - B\left(1 - t^k\right)\right)\right)\right\|_2^2 + \frac{\sigma_1}{2}\left\|\nabla t^k - v^{k+1} + \frac{s_1^k}{\sigma_1}\right\|_2^2 \quad (20)$$

$$J^{k+1} = \underset{J^k}{\operatorname{argmin}} \frac{\sigma_2}{2}\left\|J^k - w^{k+1} + \frac{s_2^k}{\sigma_2}\right\|_2^2 \quad (21)$$

$$s_1^{k+1} = s_1^k + \sigma_2\left(v^{k+1} - \nabla t^{k+1}\right) \quad (22)$$

$$s_2^{k+1} = s_2^k + \sigma_2\left(w^{k+1} - J^{k+1}\right) \quad (23)$$

The solution of (18–21) can be described as

$$v^{k+1} = \frac{\nabla t^k + \frac{s_1^k}{\sigma_1}}{\left|\nabla t^k + \frac{s_1^k}{\sigma_1}\right|} \max\left(\left|\nabla t^k + \frac{s_1^k}{\sigma_1}\right| - \frac{\nabla \varphi\left(\left|\nabla t^k\right|\right)}{\sigma_1}, 0\right) \tag{24}$$

$$w^{k+1} = \frac{\lambda_1^k t^k \Delta^{-1}\left(I - B\left(1 - t^k\right)\right) - \lambda_2^k \Delta^{-1} I_{CLAHE} + \sigma_2 J^k - s_2^k}{\lambda_1^k \left(t^k\right)^2 \Delta^{-1} - \lambda_2^k \Delta^{-1} + \sigma_2} \tag{25}$$

$$t^k = F^{-1}\left(\frac{\lambda_1^k F(\Delta^{-1}) \circ F(I) - \lambda_1^k BF(\Delta^{-1}) - \sigma_1 F(\nabla^*) \circ F\left(v^{k+1}\right) + F(\nabla^*) \circ F\left(s_1^k\right)}{\lambda_1^k F(\Delta^{-1}) \circ F\left(w^{k+1}\right) - \lambda_1^k BF(\Delta^{-1}) - \sigma_1 F(\nabla^T) \circ F(\nabla)}\right) \tag{26}$$

$$J^{k+1} = \frac{w^{k+1}\sigma_2 - s_2^k}{\sigma_2} \tag{27}$$

where $F(\ )$ is the discrete Fourier transform, $F^{-1}(\ )$ is the inverse discrete Fourier transform, * is the complex conjugate operator, and $\circ$ is the component-wise multiplication operator.

### 2.5. Red Channel Compensation Based on TEO

In order to correct the color of the underwater images and compensate for the loss of the red channel, the transmission map and background light should be estimated accurately. According to [23], the relationship between the transmission map of the red channel and the blue-green channel can be described as

$$t_b(x) = t_r(x)^{\frac{\beta_b}{\beta_r}} = t_r(x)^{\frac{B_r(-0.00113\lambda_b + 1.62517)}{B_b(-0.00113\lambda_r + 1.62517)}} \tag{28}$$

$$t_g(x) = t_r(x)^{\frac{\beta_g}{\beta_r}} = t_r(x)^{\frac{B_r(-0.00113\lambda_g + 1.62517)}{B_g(-0.00113\lambda_r + 1.62517)}} \tag{29}$$

where $B_r$, $B_g$ and $B_b$ are the background light of the red, green and blue channel, respectively. $\lambda_r$, $\lambda_g$ and $\lambda_b$ are the wavelengths of red, green and blue light. According to (28) and (29), the objective function of TEO can be defined as follows

$$f_{red}(t_r) = \left\|t_r - t_b^{\frac{B_b(m\lambda_r + i)}{B_r(m\lambda_b + i)}}\right\|_2^2 + \left\|t_r - t_g^{\frac{B_g(m\lambda_r + i)}{B_r(m\lambda_g + i)}}\right\|_2^2 \tag{30}$$

$$t_r = 1 - \min_{y \in \Omega}\left(\frac{I_r}{B_r}\right)$$

where $I_r$ is the red channel of the original underwater image. From (30), the $B_r$ and $t_r$ can be estimated by TEO. The $t_r$ is estimated by $t_g$ and $t_b$, which are optimized by adaptive non-convex non-smooth variation. $B_r$ is searched through TEO to calculate the value of (30). When the minimum of (30) is obtained, the optimal $B_r$ and $t_r$ can be estimated by TEO.

## 3. Results

In this section, the effectiveness of the proposed algorithm, compared with the state-of-the-art algorithms, such as [9,14,17,18,40], and CLAHE, is demonstrated. Both [17] and [18] are the effective IFM-based methods. The methods of [9,40] and CLAHE are typical IFM-free methods. The method of [14] is an advanced deep-learning-based method. To ensure fairness, all experiments are performed on a Windows 10 platform with an Intel Core i7-10300H central processing unit (CPU) at 2.5 GHz and with 8 GB of memory.

### 3.1. Metrics

It is non-trivial to evaluate the results of the method in datasets without ground truth, and individual metrics cannot indicate the superiority of the proposed algorithm perfectly, e.g., BRISQUE [41] measures the visual effect of the image, but it cannot indicate the restoration degree of algorithm results. In this section, BRISQUE, UIQM [42], FADE [43], UCIQE [44], and NIQE [45] are used to evaluate the restored images. UIQM is a no-reference underwater image quality metric which combines and measures the colorfulness, sharpness, and contrast of restored images. FADE evaluates the image quality via measures of the darkness pixel in underwater images. UCIQUE is a state-of-the-art method to estimate underwater image quality. NIQE is an efficient image quality analyzer based on the generalized Gaussian model. All codes of metrics are open, and the parameters of metrics are at default.

### 3.2. Qualitative Analysis

Figure 2 shows the results from different enhancement algorithms, and these images demonstrate the proposed algorithm can generate high-quality images which highlight vital visual features of the original images. The IFM-based methods, [17] and [18], can improve visual effects in some underwater images, but they may generate halo and pseudo edges, which impacts the detection of underwater resources. The IFM-free methods are able to stretch contrast and highlight the main object in underwater images. However, underwater image textures and features can be concealed due to over-enhancement and lighting over-correction. The restored images of the deep-learning-based method [14] are reddish. The proposed algorithm can output high-contrast and color saturation restored images, and it can eliminate blueish or greenish underwater scenes due to an IFM-based method and red channel compensation.

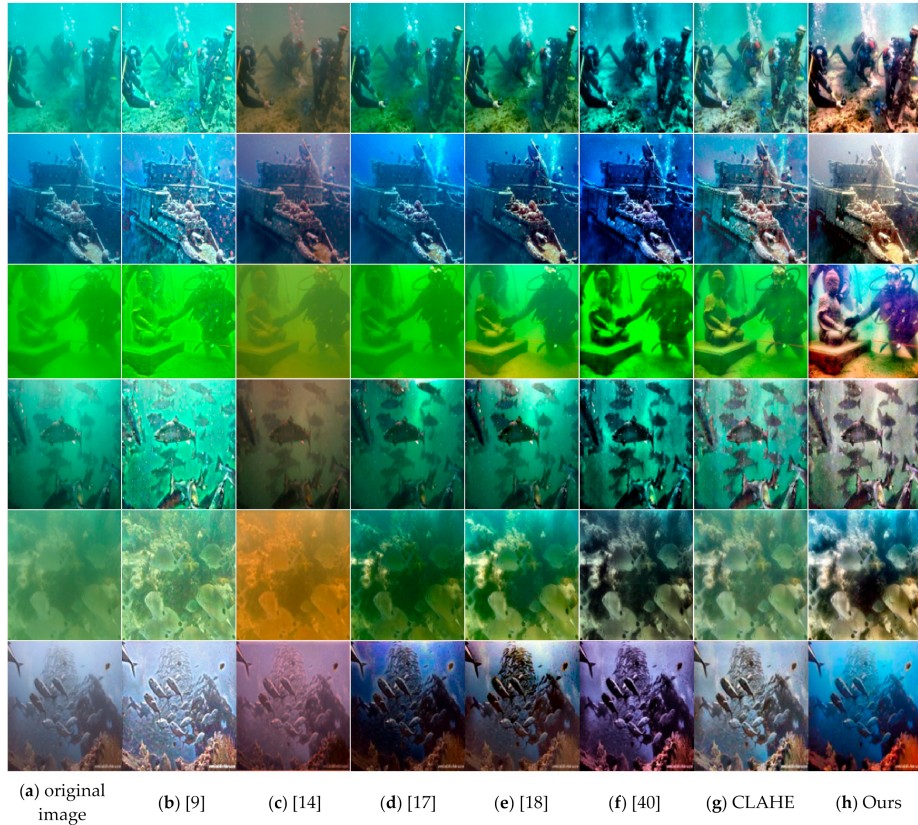

(**a**) original image　(**b**) [9]　(**c**) [14]　(**d**) [17]　(**e**) [18]　(**f**) [40]　(**g**) CLAHE　(**h**) Ours

**Figure 2.** Underwater image enhancement result. (**a**) The original image; (**b**) the result of [9]; (**c**) the result of [14]; (**d**) the result of [17]; (**e**) the result of [18]; (**f**) the result of [40]; (**g**) the result of CLAHE; (**h**) the result of ours.

### 3.3. Quantitative Analysis

The score of the metrics can be seen below. The lower scores of BRISQUE, FADE, and NIQE indicate that the image quality is better. In addition, the higher scores of UIQM and UCIQE also indicate that the image quality is better. (Figures 3–7)

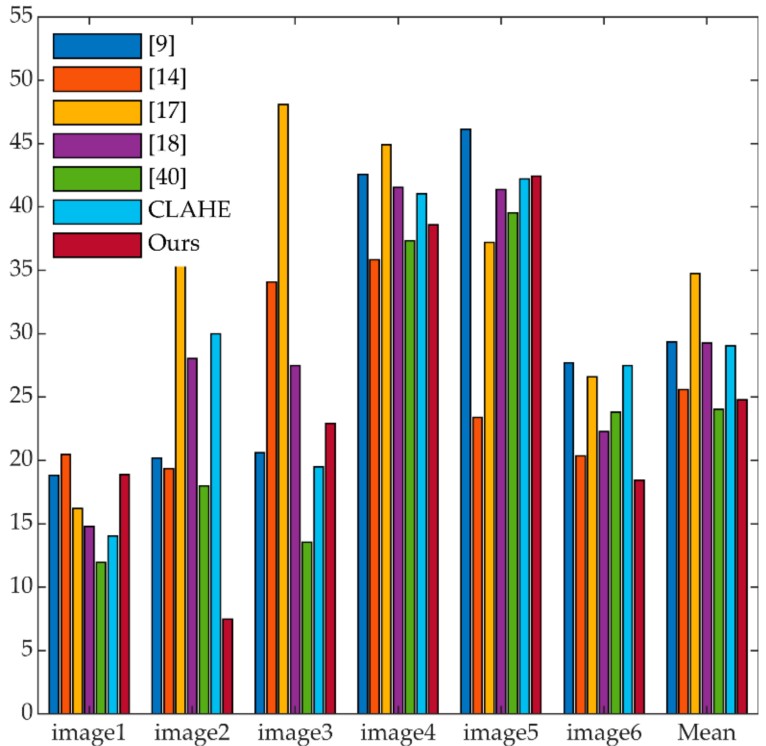

**Figure 3.** BRISQUE of restored images. A lower BRISQUE score indicates better image quality.

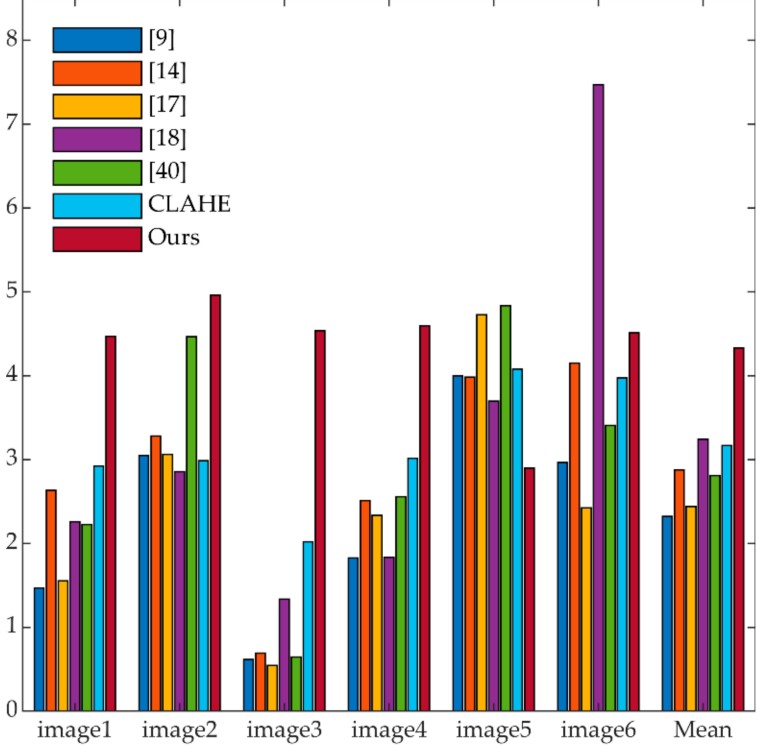

**Figure 4.** UIQM of restored images. A higher UIQM score indicates better image quality.

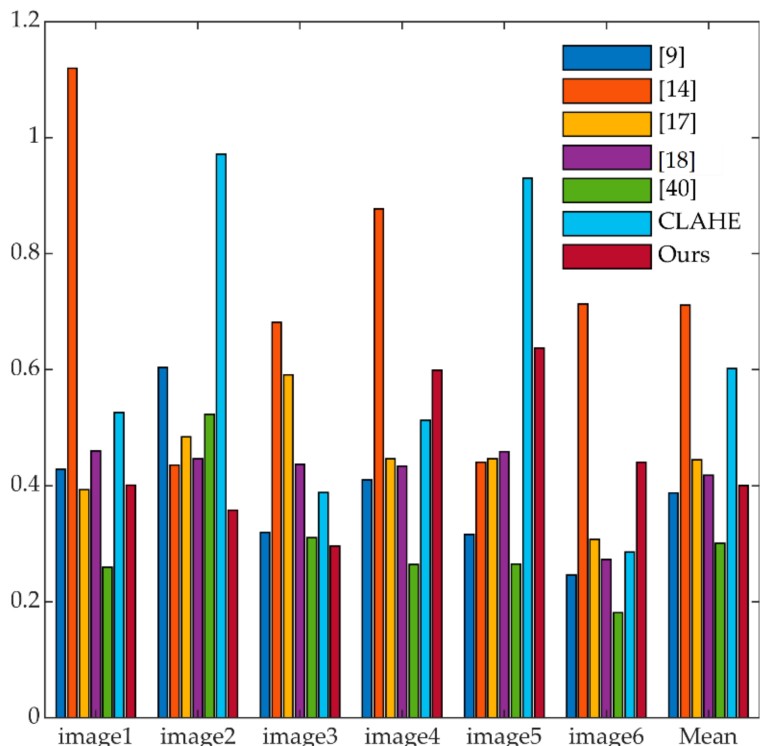

**Figure 5.** FADE of restored images. A lower FADE score indicates better image quality.

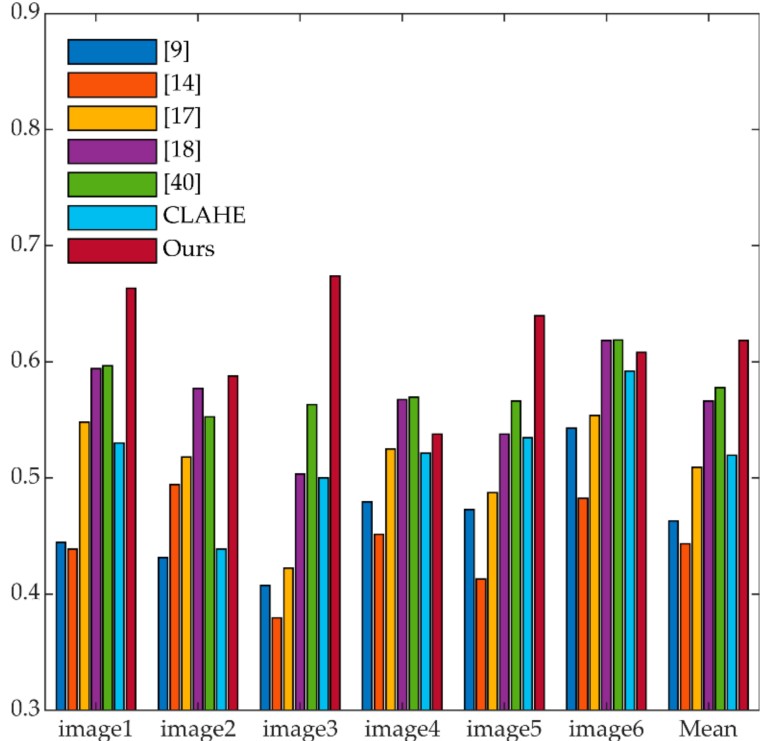

**Figure 6.** UCIQE of restored images. A higher UCIQE score indicates better image quality.

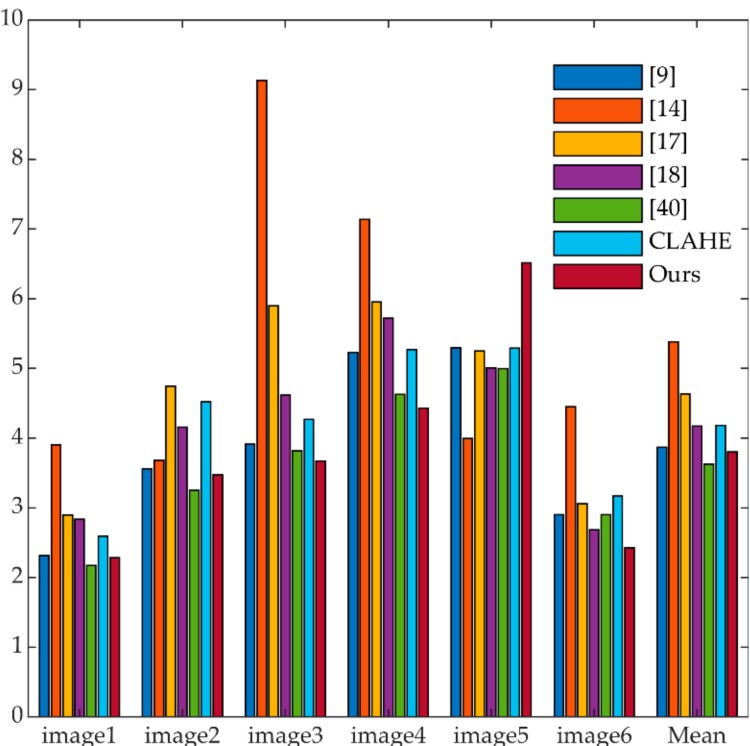

**Figure 7.** NIQE of restored images. A lower NIQE score indicates better image quality.

According to the score of BRISQUE, the proposed algorithm is the second-best method, and is close to the best method. For UIQM, the performance of the proposed algorithm scored best. According to FADE, the proposed algorithm does not perform well, because FADE measures the darkness pixels. The proposed algorithm is an IFM-based method whose stretching ability is poorer than IFM-free methods. The proposed algorithm obtained the best score from UCIQE. According to NIQE, the proposed algorithm is not the best method, but it is the second best method. In short, the proposed algorithm outputs the best-restored images, and it achieves a favorable performance compared to other methods.

*3.4. Runtime Analysis*

Running time is one of the metrics used to evaluate the practicability of underwater image restoration methods. In this part, we convert the size of the original image in Figure 2 to $100 \times 100$, $200 \times 200$, $300 \times 300$, $400 \times 400$, and $500 \times 500$, respectively, and then run them several times to obtain the running time. The running time is shown in Figure 8.

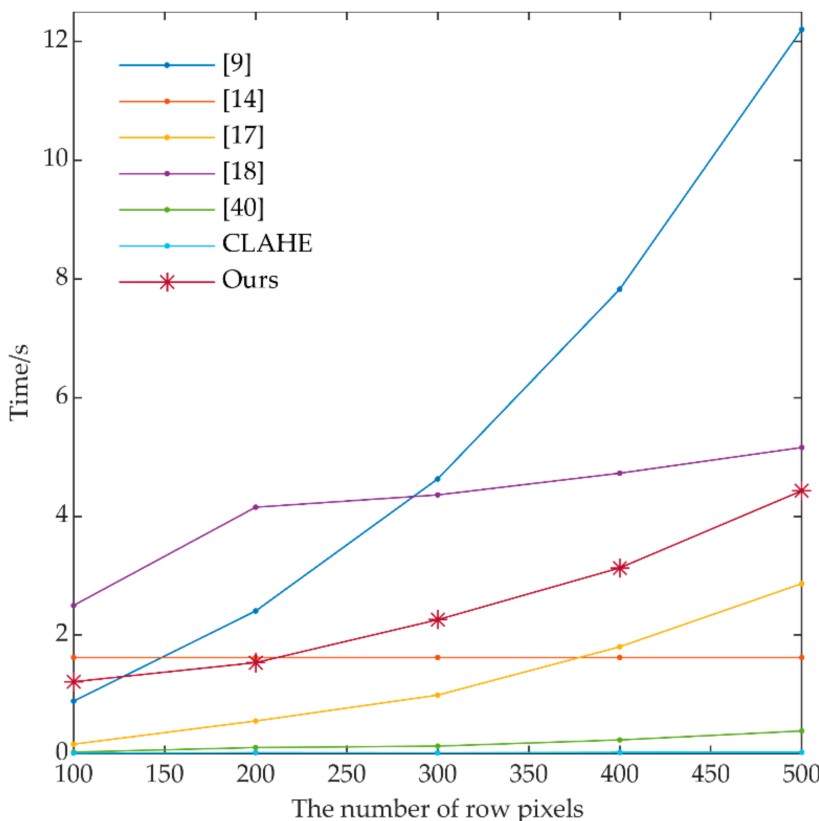

**Figure 8.** Runtime of underwater image restored methods.

According to Figure 8, the runtime of CLAHE is the quickest, and it is almost constant. The proposed method is not the best, because the non-convex non-smooth variational and TEO increase the complexity and runtime, which makes the runtime of the proposed method lower than [17].

## 4. Discussion

From qualitative and quantitative analysis, the proposed algorithm outperforms the current state-of-the-art methods in terms of restoration effect and visual effect. According to BRISQUE, FADE, and NIQE, the output of the proposed algorithm is not the best output. However, when combining the restored images, the proposed algorithm can still be considered as a better underwater image restoration method than the other alternatives.

## 5. Conclusions

In this paper, a novel underwater restoration algorithm based on DCP and non-convex non-smooth variation is proposed. Specifically, DCP is used to estimate the background light and rough transmission maps of the green and red channels. These rough transmission maps are optimized via the proposed adaptive non-convex, non-smooth variation. To compensate for the red channel, a novel red channel compensation method is proposed. Experimental results show that the proposed algorithm can obtain better results than other state-of-the-art methods.

Future developments should focus on two aspects. First, the novel variation model, which enhances edges and textures in underwater images, should be studied. Second, a rapidly converging TEO, which reduces runtime, should also be a key focus of future work.

**Author Contributions:** All authors contributed substantially to this study. Individual contributions were: Conceptualization, M.L. and Y.Z.; methodology, Q.J., P.L., and M.L.; software, Q.J., M.H. and

L.K.; validation, Q.J., M.L. and L.D.; formal analysis, Q.J.; investigation, P.L. and M.L.; resources, M.L., L.D., and Y.Z.; data curation, Q.J. writing—original draft preparation, Q.J.; writing—review and editing, Q.J. and P.L.; visualization, Q.J.; supervision, M.L. and Y.Z.; project administration, M.L.; funding acquisition, M.L. All authors have read and agreed to the published version of the manuscript.

**Funding:** This research was funded by the national key research and development project of China, grant number 2018YFF0300804.

**Institutional Review Board Statement:** Not applicable.

**Informed Consent Statement:** Not applicable.

**Data Availability Statement:** The publicly archived datasets of the underwater image data used in this paper, are derived from website: https://li-chongyi.github.io/proj_benchmark.html (accessed on 23 April 2021) and https://github.com/dlut-dimt/RealworldUnderwater-Image-Enhancement-RUIE-Benchmark (accessed on 23 April 2021).

**Conflicts of Interest:** The authors declare no conflict of interest.

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
