# Peer review of "Underwater Image Restoration via Non-Convex Non-Smooth Variation and Thermal Exchange Optimization"

_jmse, doi:10.3390/jmse9060570_

Round 1

Reviewer 1 Report

1. Introduction - Maybe you could consider addressing also:
10.1109/TIP.2017.2663846
https://doi.org/10.1007/s11042-020-10049-7
10.1109/ACCESS.2018.2875344
https://www.sciencedirect.com/science/article/pii/S0030399219306395

2. Used metrics BRISQUE, UIQM, FADE, UCIQE, and NIQE should be defined in the paper in form of expression. Even more, references for metrics should be also cited. 

3. Lines 107-111 are not well elaborated. Why are you using such procedure. G and B channel are used to optimized R channel. Specific reason for such procedure should be in the article text. Is it possible to optimize G form R and B? Is it your choice or the result form your research? If you know this from the reference (that R should be optimized from other 2 channels) that the reference should be cited.

Author Response

Response to Reviewer 1 Comments

Dear Reviewer:

It is our pleasure to have you review our manuscript, and thank you for taking time out of your busy schedule to give us your valuable comments. The following is our modification:

Point 1: Introduction - Maybe you could consider addressing also:

(1)10.1109/TIP.2017.2663846

(2)10.1007/s11042-020-10049-7

(3)10.1109/ACCESS.2018.2875344

(4)https://www.sciencedirect.com/science/article/pii/S0030399219306395

Response 1:

Thank you for your valuable comments, which have improved our manuscript.

We have added your suggested references to our manuscript.

(1) “Peng et al. consider the problem from image blurriness and light absorption, and pro-pose a depth estimation method to obtain restored image [19]”

(2) “In the optimization of transmission map, such as guided filtering [25, 26]”

(3) “Zhang et al. improved IMF, and estimate the medium transmissions of three channels of an underwater image via joint prior [20].”

(4) “The attenuation curves prior is applied to estimate transmission maps by Dai et al. they also propose a color balance algorithm estimate restored image with natural appearance [21].”

(5) “Contrastive learning and generative adversarial networks are utilized by Han et al to maximize mutual information between dataset [16].”

Considering the achievements of unsupervised learning and self-supervised learning in deep learning, we have added additional references, which can complement the you suggested references (5) effectively, and enriches the summary of underwater image restoration based on deep learning.

Recently, unsupervised and self-supervised learning have become a hot topic in deep learning, and they have achieved excellent performance in underwater image restoration. Ye et al. built an unsupervised adaptation network to estimate the depth map and restored image [15]”

http://doi.org/10.1109/TCSVT.2019.2958950

Point 2:  Used metrics BRISQUE, UIQM, FADE, UCIQE, and NIQE should be defined in the paper in form of expression. Even more, references for metrics should be also cited.

Response 2:

Thank you for your valuable comments.

We have added references to these metrics, and stating that all parameters of metrics are set by the default parameters in the code.

BRISQUE [41], UIQM [42], FADE [43], UCIQE [44], and NIQE [45]

Point 3:  Lines 107-111 are not well elaborated. Why are you using such procedure. G and B channel are used to optimized R channel. Specific reason for such procedure should be in the article text. Is it possible to optimize G form R and B? Is it your choice or the result form your research? If you know this from the reference (that R should be optimized from other 2 channels) that the reference should be cited.

Response 3:

Thank you for your valuable comments.

According to the underwater imaging mechanism, the attenuation of absorption and scattering for red light is the fastest, there are errors by using the most attenuation channel (red channel) to estimate the transmission map of other channels.

According to previous work, Azmi et al. used the blue and green channels to estimate the red channel, and then the particle swarm optimization is used to perform contrast stretching [22]. Galdran et al. employed the method of red channel compensation based on DCP, which realized color correction and visibility improvement [24]

In conclusion, we use the Green and Blue channel to estimate Red channel, and above supplement has been added to our manuscript:

Because the attenuation of absorption and scattering for red light is the fastest, there are errors by using the most attenuation channel (red channel) to estimate the transmission map of other channels.

Reviewer 2 Report

The paper is well-written and well-presented. Qualitatively speaking, the results achieved by this method are visually appealing. Additionally, the method is quantifiably evaluated (insofar as possible) using a range of metrics.

The method is compared against other IMF-free and IMF-based methods. It would be nice to see results from a deep learning based approach as well (e.g. UWGAN - Underwater GAN), but this is just a suggestion. I'll leave it to the discretion of the authors if they wish to include this.

It would be nice to know some of the details around computation speed for this method.

I think Figure 1 could benefit from being reworked. I don't see where Thermal Exchange Optimisation is represented. Also, there are arrowheads at both ends of the line between 'Red transmission map' and 'Red background light'. I found this figure a bit hard to understand.

Some minor grammar/issues are present in the manuscript but, in general, the English is good.

Author Response

Response for reviewer 2:

Dear Reviewer:

It is our pleasure to have you review our manuscript, and thank you for taking time out of your busy schedule to give us your valuable comments. The following is our modification:

Point 1: The method is compared against other IMF-free and IMF-based methods. It would be nice to see results from a deep learning based approach as well (e.g. UWGAN - Underwater GAN), but this is just a suggestion. I'll leave it to the discretion of the authors if they wish to include this.

Response 1:

Thank you for your valuable comments, which have improved our manuscript.

We have supplemented the experiment by using Li et al. proposed method, which is based on deep learning, and they code can be downloaded in Github. we have added the restored images via Li et al to our manuscript in Figure 2 – 7.

In addition, we have added this work to the Introduction section.

http://doi.org/10.1016/j.patcog.2019.107038

Thank your comments to improve our Introduction and Experiment

Point 2: It would be nice to know some of the details around computation speed for this method.

Response 2:

Thank you for your valuable comments.

We have supplemented the experiment, which can be seen in 3.4 runtime analysis. In this experiment, We changed the size of the original image in Figure 2 to 100×100,200×200, 300×300, 400×400, and 500×500. The experimental results are shown in our manuscript (Figure 8).

Point 3: I think Figure 1 could benefit from being reworked. I don't see where Thermal Exchange Optimization is represented. Also, there are arrowheads at both ends of the line between 'Red transmission map' and 'Red background light'. I found this figure a bit hard to understand.

Response 3:

Thank you for your valuable comments.

We have reworked Figure 1 based on your comments and the work of others, and it can be seen in our manuscript as Figure 1:

Point 4: Some minor grammar/issues are present in the manuscript.

Response 4:

Thank you for your valuable comments.

We checked and corrected some of the grammar/issues as best we could. The details can be seen in our manuscript.
